# Presepsin in the Rapid Response System for Cancer Patients: A Retrospective Analysis

**DOI:** 10.3390/jcm10102153

**Published:** 2021-05-16

**Authors:** Min-Jung Lee, Won-Ho Han, June-Young Chun, Sun-Young Kim, Jee-Hee Kim

**Affiliations:** 1National Cancer Center, Department of Surgery, Goyang 410-769, Korea; 63288@ncc.re.kr (M.-J.L.); 13408@ncc.re.kr (W.-H.H.); 2National Cancer Center, Department of Internal Medicine, Goyang 410-769, Korea; june.y.chun@ncc.re.kr; 3National Cancer Center, Department of Cancer Control and Population Health, Graduate School of Cancer Science and Policy, Goyang 410-769, Korea; sykim@ncc.re.kr; 4National Cancer Center, Department of Anesthesiology, Goyang 410-769, Korea

**Keywords:** presepsin, septic shock, rapid response system, diagnosis, lactic acid

## Abstract

Introduction: Early diagnosis of sepsis is paramount to effective management. The present study aimed to compare the prognostic accuracy of presepsin levels and other biomarkers in the assessment of septic shock and mortality risk in cancer patients. Materials and methods: A total of 74 cancer patients were evaluated for presepsin, lactic acid, C-reactive protein (CRP) levels, and white blood cell count (WBC). Specificity and sensitivity values for septic shock and death were compared between four biomarkers in all patients and those with and without acute kidney injury (AKI). Results: A total of 27 and 29 patients experienced septic shock and died, respectively. The area under the curve (AUC) and sensitivity and specificity estimated for presepsin levels for septic shock were 60%, 74%, and 51%, respectively. The corresponding values for mortality were 62%, 72%, and 49%, respectively. In patients without AKI, AUC of presepsin levels for septic shock and death were 62% and 65%, respectively; in those with AKI, these values were 44% and 58%, respectively. Presepsin levels showed higher sensitivity and specificity values than WBC and higher specificity than CRP but were similar to those of lactic acid levels. Conclusions: Presepsin levels are similar to lactic acid levels in the assessment of septic shock and mortality risk in cancer patients. In patients with AKI, presepsin levels should be considered carefully.

## 1. Introduction

Sepsis is associated with high rates of morbidity and mortality, in particular, among critically ill patients. Early diagnosis and treatment of sepsis are paramount to preventing septic shock and multiorgan failure. As a result, clinicians tend to prioritize early diagnosis of sepsis and stratification of patients at risk [1].

In its early stages, sepsis involves the host’s immune responses, which lead to the amplification of proinflammatory activity as well as that of responses against microorganisms [2]. CD14 is a glycoprotein expressed in macrophage, monocyte, and granulocyte cells that plays an important role in innate immunity. It includes a receptor with a high affinity for lipopolysaccharide/lipopolysaccharide binding protein (LPS/LPB) complexes. After LPS/LPB complex binds with CD14, soluble CD14 subtype, also known as presepsin, is cleaved from this complex and released into circulation [3,4,5].

Presepsin levels have been reported to increase in patients with sepsis and to be indicative of sepsis severity in some of them. It tends to be considered as a prominent biomarker in the diagnosis of sepsis or cancer from recent studies [5,6]. Meanwhile, other studies have reported that presepsin is eliminated through the kidneys, suggesting that patients with renal failure and at risk of sepsis should be monitored carefully. Nevertheless, interpretation of such findings in patients with renal failure remains a challenge [7].

Rapid response systems (RRS) screen patients at risk of cardiac arrest in hospital. Early detection and sepsis severity evaluation are particularly important in rapid response systems; however, no previous study has reported the use of presepsin levels by RRS. This study aimed to evaluate the early diagnostic and prognostic value of serum presepsin levels compared to that of inflammatory markers, such as procalcitonin and lactic acid levels, and white blood cell count (WBC) in patients who had experienced septic shock or died.

## 2. Materials and Methods

### 2.1. Study Design and Participants 

This retrospective observational study was conducted at a single tertiary hospital, the National Cancer Center in Korea. Data from a total of 106 cancer patients suspected of having sepsis by RRS screening between March and April 2020 were included. Blood samples were evaluated for presepsin levels. Patients lacking data on lactic acid or C-reactive protein (CRP) levels were excluded. Finally, a total of 74 patients were included. 

Participants were dichotomized based on sepsis and septic shock groups, following the Third International Consensus Definitions for Sepsis and Septic Shock [8]. Patients were categorized into the septic shock group if they met at least one of the following criteria: respiratory failure with the ratio of the partial pressure of oxygen to the fraction of inspired oxygen <250, renal failure with urine output <0.5 mL/kg/h, hematological organ failure with platelet count, and cardiovascular organ failure requiring vasopressors for at least 1 day. Mortality was defined as death within 30 days after presepsin-level evaluation. At the time of presepsin-level evaluation, an increase in serum creatinine levels ≥0.3 mg/dL within 48 h or in urine volume <0.5 mL/kg/h for 6 h was defined as indicative of acute kidney injury (AKI) [9]. This study was approved by the Institutional Review Board of the National Cancer Center (approval no. NCC 2021-0151). Goyang, Korea.

### 2.2. Statistical Analyses

The optimal cutoff value of presepsin levels was determined as the point with high specificity and sensitivity in receiver operating characteristic (ROC) curve analysis. The relationships of each of the four inflammatory markers, including CRP, lactic acid, WBC, and presepsin with mortality or septic shock, were evaluated by using Pearson’s chi-square test. Sensitivity and specificity of all inflammatory markers were examined as indicators of severity of sepsis and mortality risk. These diagnostic accuracy indicators of presepsin for mortality and septic shock were compared with those of CRP, lactic acid, and WBC. The diagnostic accuracy was also compared in patients with and without AKI by using re-identified cutoff of presepsin levels. The area under the curve (AUC) was calculated to examine sensitivity and specificity of all inflammatory markers as indicators of severity of sepsis and mortality risk. The *p*-values of <0.05 were considered indicative of a statistically significant finding. 

## 3. Results

### 3.1. Patient Demographics

As a result of RRS activation, 86.5% of the patients were transferred to the ICU with the diagnosis of sepsis or septic shock in 17 (23.0%) and in 27 (36.5%) patients, respectively. Hepatobiliary and pancreatic cancer was the most common cancer type (*n*= 19, 25.6%), followed by gastrointestinal (*n* = 14, 18.9%), lung cancer (*n* = 10, 13.5%), and 21 patients (28.4%) had metastatic cancer. A total of 53 (71.6%) patients underwent chemotherapy within 1 month. In patients with continuous vasoactive agent, mechanical ventilator care, and continuous renal replacement therapy, 16 (16/44 36.3%), 14 (14/21 66.6%), and 6 (6/14 42.8%) died, respectively. There was a high correlation with the mortality rate in mechanical ventilator care. Finally, 29 (39.2%) patients died within 30 days (Table 1).

### 3.2. Markers of Septic Shock and Mortality Risk

The histogram of four biomarkers is depicted in Figure 1. 

The ROC curve analysis revealed that the cutoff values of presepsin levels predictive of septic shock and death were 728 pg/mL (area under the curve [AUC], sensitivity, and specificity values of 60%, 74%, and 51%, respectively) and 727 pg/mL (62%, 72%, 49%, respectively) (Figure 2). 

Lactic acid levels were positively related with septic shock (*p* = 0.0443) and mortality (*p* = 0.0129) risk. Although average presepsin levels were higher in the deceased and those with septic shock, the relationships between presepsin levels and either septic shock or death were not statistically significant (Figure 3, Table 2). Sensitivity and specificity of presepsin levels were generally higher than those of WBC (59% and 40% for septic shock; 52% and 36% for mortality) or CRP levels (96% and 6%; 97% and 7%) but similar to those of lactic acid levels (70% and 57%; 69% and 58%) (Table 3). When classified by sequential organ failure assessment (SOFA) score, presepsin level showed positive correlation with mortality rate as SOFA score (Figure 4).

### 3.3. Markers of Septic Shock in Patients with/without Acute Kidney Injury

In patients without AKI, AUC values for septic shock and death were 62% and 65% with the presepsin cutoff values of 799 and 727 pg/mL, respectively. The corresponding values for patients with AKI were 44% and 58% and 1000 and 1630 pg/mL, respectively. The cutoff values for patients with AKI were higher than those for patients without AKI (Figure 5, Table 4).

## 4. Discussion

Early detection of septic shock is important in cancer patients who are immuno-compromised due to a decrease in performance status or chemotherapy. Previous studies have reported that presepsin levels may be helpful in the early diagnosis of sepsis; however, their usefulness to RRS remains unclear. The present study aimed to evaluate the effectiveness of presepsin levels as a diagnostic and prognostic marker for early detection of septic shock in cancer patients. To the best of our knowledge, the present study is the first to report findings from the analysis of serum presepsin levels in patients treated at a cancer-specific hospital that uses RRS monitoring. The present findings suggest that presepsin levels might be a sensitive biomarker of septic shock in cancer patients. 

RRS is used to monitor patients considered at risk of needing an early intervention. Several biomarkers should be considered in early diagnosis of sepsis. WBC and CRP levels tend to be used in this context; however, WBC was associated with low sensitivity and specificity in the present study, while CRP levels might not be suitable as a diagnostic marker in early detection of sepsis [10,11]. Presepsin is a fragment of the N-terminal sequence of soluble CD14 occurring in the early inflammatory phase; its levels are known to increase more rapidly than those of CRP, procalcitonin, or interleukin-6 [6,7]. It has been suggested as useful in the diagnosis of septic shock at bedside (alongside lactic acid levels); it is currently used in clinical practice in this context.

Presepsin levels have been reported as a useful biomarker of sepsis in the emergency department or ICU [12,13]. Previous studies have shown that presepsin levels increase in the early stages of sepsis, representing an innate immunity response, and that they positively correlate with the severity of sepsis. In the present study, the cutoff values for presepsin levels indicative of septic shock and mortality risk were 728 pg/mL (AUC, sensitivity, and specificity of 60%, 74%, 51%, respectively), and 727 pg/mL (AUC, sensitivity, and specificity of 62%, 72%, and 49%, respectively). Previous studies have reported presepsin-level cutoff values for sepsis in the range from 500 to 1000 pg/mg. In the present study, the cutoff values were relatively high, likely due to the fact that we used septic shock as the outcome of interest. 

Several inflammatory markers are influenced by the patient’s condition and the underlying disease. WBC is affected by hematologic malignancy, chemotherapy, or renal function. [14] Lactic acid level was reported to be increased in malignancy or chemotherapy associated with anaerobic metabolism. [15,16] However, the relationship between kidney function and lactic acid level has not been reported yet. In this study, lactic acid has higher sensitivity and specificity than presepsin in patients with AKI, suggesting that kidney function has relatively little effect on lactic acid. Procalcitonin, which is another biomarker of sepsis, is also affected by renal function [17,18]. Presepsin levels were high in patients with AKI; this finding is consistent with that of a previous study [7]. This finding suggests that presepsin levels should be interpreted with caution in the diagnosis and prognosis of sepsis. Because sepsis is the most common cause of AKI, such biomarkers should be interpreted with caution when diagnosing sepsis in patients with AKI. 

Early detection of sepsis is crucial to improve survival rate in sepsis with malignancy [19,20]. Medical staff in rapid response systems often need to differentiate sepsis in the early stages of clinical deterioration. This is one of the difficult aspects of the rapid response system [21]; presepsin could be a useful marker in diagnosing septic shock. Presepsin is helpful not only in diagnosis of septic shock but also as an indicator for predicting mortality, along with lactic acid and SOFA scores. When AKI is accompanied, it may be measured higher than in patients without AKI; it is necessary to pay attention to interpretation and consider other markers to diagnosis septic shock or predict mortality.

This study has several limitations. First, the sample size was small. Second, this was a retrospective, single-center study, which may have affected the presented estimates. Third, in contrast to some other studies, where presepsin levels were measured daily over the course of several days, in the present study, these levels were measured only once. 

## 5. Conclusions

In cancer patients, presepsin levels may be more suitable for use as a biomarker of septic shock and death risk than are lactic acid levels; however, they should be used with caution in patients with AKI. 

## Figures and Tables

**Figure 1 jcm-10-02153-f001:**
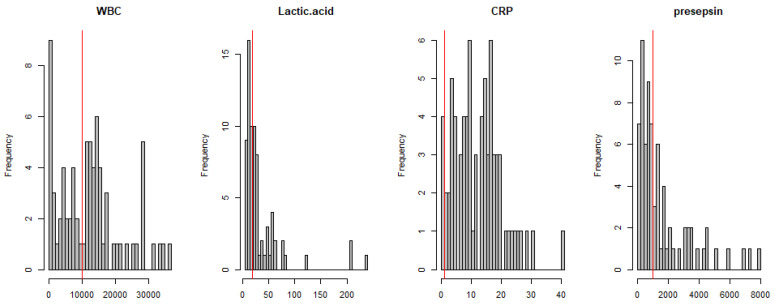
Histograms of laboratory markers.

**Figure 2 jcm-10-02153-f002:**
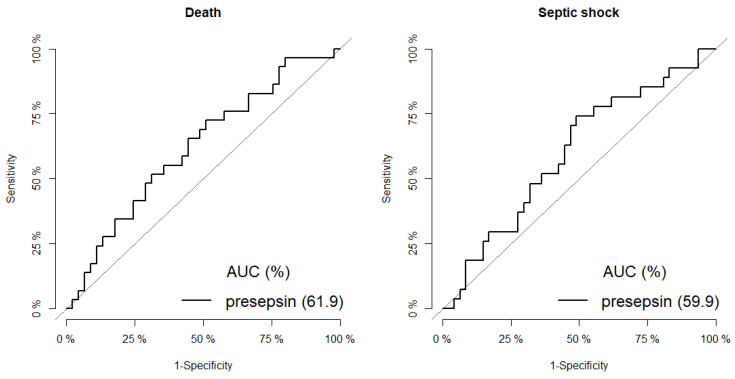
ROC curve by death (**left**) and septic shock (**right**).

**Figure 3 jcm-10-02153-f003:**
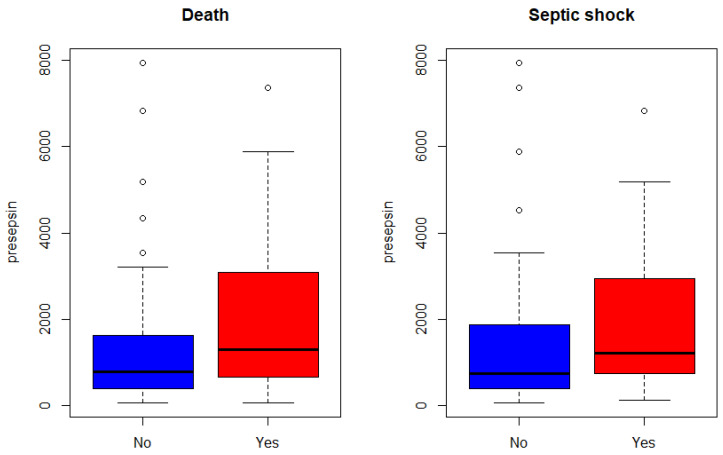
Box plots of presepsin by death and septic shock. Blue bar means no event (no patient death or septic shock); red bar means event (death or septic shock).

**Figure 4 jcm-10-02153-f004:**
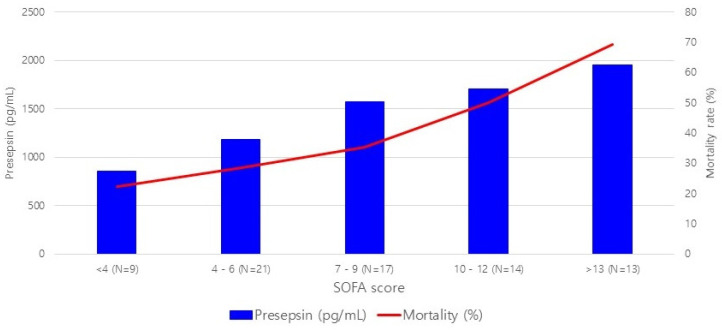
Correlation of presepsin level and SOFA score with mortality.

**Figure 5 jcm-10-02153-f005:**
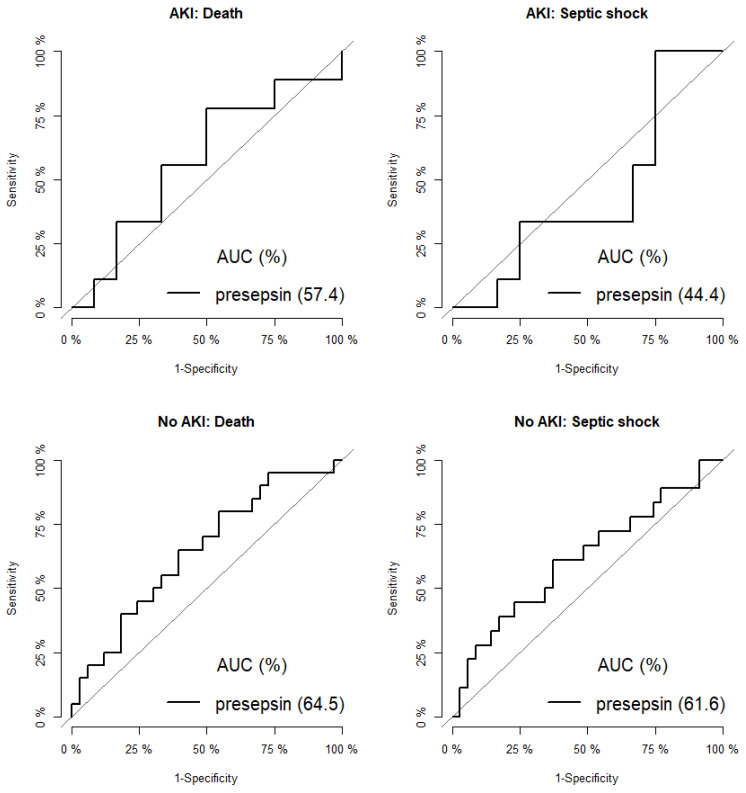
ROC curves by mortality (**left**) and septic shock (**right**) with/without acute kidney injury.

**Table 1 jcm-10-02153-t001:** Patient characteristics.

Variable	Number of Patients (%) (*n* = 74)
Sex	
Male	41 (55.4%)
Female	33 (44.6%)
Age	64.6 (13.6%)
Cancer type	
Hepatobiliary and pancreatic cancer	19 (25.6%)
Gastrointestinal cancer	14 (18.9%)
Gynecological cancer	11 (14.8%)
Urological cancer	10 (13.5%)
Lung cancer	10 (13.5%)
Hematologic malignancy	6 (8.1%)
Others	4 (4.9%)
Cancer stage	
Locally advanced cancer	53 (71.6%)
Metastatic cancer	21 (28.4%)
ICU transfer	64 (86.5%)
Yes	10 (13.5%)
No	
Diagnosis at RRS activation	
Sepsis	17 (23.0%)
Septic shock	27 (36.5%)
Other shock	21 (28.4%)
Respiratory failure	7 (9.5%)
Others	2 (2.7%)
Pre-existing condition	
Chemotherapy within 30 days	53 (71.6%)
Cardiovascular disease	10 (13.5%)
With acute kidney injury	21 (28.3%)
Intervention in place	
Continuous vasoactive agent use	44 (59.5%)
Mechanical ventilator use	21 (28.4%)
Continuous renal replacement therapy	14 (18.9%)
Mortality	29 (39.2%)

**Table 2 jcm-10-02153-t002:** Diagnostic test of laboratory markers for septic shock and mortality.

Variable	Cut off	Sensitivity (95% CI)	Specificity (95% CI)
Septic shock			
WBC	12,000	59.3	40.4
CRP	1.0	96.3	6.4
Lactic acid	19.8	70.4	57.4
Presepsin	728	74.1	51.1
Mortality			
WBC	12,000	51.7	35.6
CRP	1.0	96.6	6.7
Lactic acid	19.8	69.0	57.8
Presepsin	727	72.4	48.9

**Table 3 jcm-10-02153-t003:** Comparison of laboratory markers for septic shock and mortality.

Variable	Cut off	Non-Septic Shock (*n* = 47)	Septic Shock (*n* = 27)	*p*-Value *
Septic shock				0.918
WBC	<12,000	19	11	
	≥12,000	28	16	
CRP	<1.0	3	1	0.441
	≥1.0	44	26	
Lactic acid	<19.8	27	8	0.012
	≥19.8	20	19	
Presepsin	<728	24	7	0.079
	≥728	23	20	
Mortality				
WBC	<12,000	16	14	0.397
	≥12,000	29	15	
CRP	<1.0	3	1	1.000
	≥1.0	42	28	
Lactic acid	<19.8	26	9	0.044
	≥19.8	19	20	
Presepsin	<727	22	8	0.114
	≥727	23	21	

* Chi-square test.

**Table 4 jcm-10-02153-t004:** Diagnostic test of laboratory markers for septic shock with/without acute kidney injury.

Variable	Normal Kidney Function		With AKI	
	Sensitivity (95% CI)	Specificity (95% CI)	Sensitivity (95% CI)	Specificity (95% CI)
Septic shock				
Lactic acid	55.6 (30.8–78.5)	60 (42.1–76.1)	100 (66.4–100)	50 (21.1–78.9
Presepsin	61.1 (35.7–82.7)	62.9 (44.9–78.5)	100 (66.4–100)	25 (5.5–57.2)
Mortality				
Lactic acid	60 (36.1–80.9)	63.6 (45.1–79.6)	88.9 (51.8–99.7)	41.7 (15.2–72.3)
Presepsin	65 (40.8–84.6)	60.6 (42.1–97.2)	77.8 (40–97.2)	50 (21.1–78.9)

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
