# Peer review of "Presepsin in the Rapid Response System for Cancer Patients: A Retrospective Analysis"

_jcm, 2021, doi:10.3390/jcm10102153_

Round 1
Reviewer 1 Report
Jung and Colleagues have performed a retrospective study on cancer patients admitted into ICU and suspected the pathogenesis of sepsis. Herein authors have extracted CRP, Lactic acid, WBC count, and presepsin and further correlate with patient mortality and septic sock and shown specificity and sensitivity of Lactic acid and presepsin in AKI under septic sock and mortality.
Presepsin is a receptor of lipopolysaccharide-lipopoly-saccharide binding protein (LPS-LBP) complexes and a known marker of sepsis diagnosis with high sensitivity and specificity.
cancer patients are prone to developing lactic acidosis. however, it quite varies between cancer types. In the current manuscript, the authors haven't provided the details of cancer patients such as cancer stage therapy etc and need to be included in this study. It is highly likely if the patient is in ICU might be on the ventilator and which could be a major factor of sepsis development. The authors haven't provided any information on whether patients were on a ventilator or not? did they were on medication/chemotherapy/antibiotic treatment etc? Metabolic acidosis and change in immune cell response are very possible in cancers and acute kidney failure. It is not clear from data whether changes in WBC, lactic acid has any independent association with AKI in sepsis or mortality or it is an artifact of cancer? Authors may consider calculating Sequential Organ Failure Assessment (SOFA) and Acute Physiology, Age and Chronic Health Evaluation (APACHE) score system and correlate with Presepsin. Authors may distinguish the patients who died on a ventilator vs off ventilator.
Figure 3 is not clear with their interpretation. Need to describe the figure under footnote.
Author Response
May 6, 2020
Manuscript ID: jcm-1207776
Title: Presepsin in the rapid response system for cancer patients: A retrospective analysis
We thank Professor Mr. Aldrich Liu and reviewers of Journal of Clinical Medicine for positive consideration on our manuscript. We would like to express our sincere gratitude to the reviewers and editor for their time and effort in reviewing our manuscript. Each and every comment has helped to improve our manuscript in various aspects. We have tried our best to answer to the questions and comments put forward by the reviewers. Thank you.
Reply to Reviewer.
Point 1. In the current manuscript, the authors haven't provided the details of cancer patients such as cancer stage therapy etc and need to be included in this study.
Response 1. Authors added type of cancer and metastatic cancer to Results according to the reviewer's opinion. We added in result as “Hepatobiliary and pancreatic cancer was the most common cancer type (n=19, 25.6%), followed by gastrointestinal (n=14, 18.9%) and lung cancer (n=10, 13.5%) and 21 patients (28.4%) had metastatic cancer.”
Point 2. It is highly likely if the patient is in ICU might be on the ventilator and which could be a major factor of sepsis development. The authors haven't provided any information on whether patients were on a ventilator or not? did they were on medication/chemotherapy/antibiotic treatment etc?
Response 2: Authors showed the patient demographics in table 1, including the number of patients with metastatic cancer, with the history of chemotherapy within 30 days and who went through the intervention (CRRT, mechanical ventilator, vasoactive agent use).
We added the number of patients on ventilator to Results according to the reviewer’s opinion,
“In patients with continuous vasoactive agent, mechanical ventilator care, and continuous renal replacement therapy, 16 (16/44 36.3%), 14 (14/21 66.6%), and 6 (6/14 42.8%) died respectively.”
Point 3. Metabolic acidosis and change in immune cell response are very possible in cancers and acute kidney failure. It is not clear from data whether changes in WBC, lactic acid has any independent association with AKI in sepsis or mortality or it is an artifact of cancer?
Response 3: Authors agree with the reviewers. However, we did not analyze the effect of WBC on AKI due to the low sensitivity and specificity of WBC to sepsis. We added in discussion as below
“Several inflammatory markers are influenced by the patient's condition and the underlying disease. WBC is affected by hematologic malignancy, chemotherapy or renal function. Lactic acid level was reported to be increased in malignancy or chemotherapy associated with anaerobic metabolism. However, the relationship between kidney function and lactic acid level has not yet been reported. In this study, lactic acid has higher sensitivity and specificity than presepsin in patients with AKI, suggesting that lactic acid has relatively little effect on kidney function. procalcitonin, which is another biomarker of sepsis, are affected by renal function.”
Point 4. Authors may consider calculating Sequential Organ Failure Assessment (SOFA) and Acute Physiology, Age and Chronic Health Evaluation (APACHE) score system and correlate with Presepsin.
Response 4: When classified by SOFA score, presepsin level showed positive correlation with mortality rate as SOFA score. We added Figure 4. correlation of presepsin level and SOFA score with mortality
Point 5. Authors may distinguish the patients who died on a ventilator vs off ventilator.
Response 5: We added the number of patients who died on a ventilator in result. “In patients with mechanical ventilator care, 14 patients (14/21 66.6%) died respectively.”
Point 6. Figure 3 is not clear with their interpretation. Need to describe the figure under footnote.
Response 6: Authors added the interpretation under footnote. “Blue bar means no event (patient not death or septic shock), red bar means event (death or septic shock).”
Thanks reviewers of Journal of Clinical Medicine.
Sincerely yours,
Jee Hee Kim
Department of Anesthesiology, National Cancer Center
323 Ilsan-ro, Ilsandong-gu, Goyang-si 410-769, Gyeonggi-do, Republic of Korea
E-mail: [email protected]

Reviewer 2 Report
In this manuscript Jung and colleagues evaluated the prognostic accuracy of presepsin, as indicator of sepsis severity, and other biomarkers in cancer patients in relation to acute kidney injury.
Presepsin role as biomarker in the diagnosis of sepsis or cancer has been reported previously. This should be discussed properly in the introduction part.
The analysis of the “area under the curve (AUC)” should be described in material and methods.
Figure 3 Please state clearly what is in red and what is in blue.
Implications of the correlation between presepsin and septic shock or death in cancer patients should be described further. Moreover, the clinical implications of the role of presepsin in case of AKI could be further discussed.
Author Response
May 6, 2021
Manuscript ID: jcm-1207776
Title: Presepsin in the rapid response system for cancer patients: A retrospective analysis
We thank Professor Mr. Aldrich Liu and reviewers of Journal of Clinical Medicine for positive consideration on our manuscript. We would like to express our sincere gratitude to the reviewers and editor for their time and effort in reviewing our manuscript. Each and every comment has helped improve our manuscript in various aspects. We have tried our best to answer to the questions and comments put forward by the reviewers. Thank you.
Reply to Reviewer.
Reviewer 2
Point 1. Presepsin role as biomarker in the diagnosis of sepsis or cancer has been reported previously. This should be discussed properly in the introduction part.
Response 1. Authors added about previous findings that presepsin increases in septic condition in the introduction part. “Presepsin levels have been reported to increase in patients with sepsis and to be indicative of sepsis severity in some of them. It tend to be considered as prominent biomarker in the diagnosis of sepsis or cancer from recent studies”
Point 2. The analysis of the “area under the curve(AUC)” should be described in material and methods.
Response 2. We added the description about AUC in material and methods. “The AUCs were calculated to examine sensitivity and specificity of all inflammatory markers as indicators of severity of sepsis and mortality risk.”
Point 3. Figure 3 Please state clearly what is in red and what is in blue
Response 3. We added the footnote about it. ”Blue bar means no event (patient not death or septic shock), red bar means event (death or septic shock).”
Point 4. Implications of the correlation between presepsin and septic shock or death in cancer patients should be described further. Moreover, the clinical implications of the role of presepsin in case of AKI could be further discussed.
Response 4. We added in Discussion as below
“Early detection of sepsis is crucial to improve survival in sepsis with malignancy. Medical staff in rapid response systems often need to differentiate sepsis in the early stages of clinical deterioration. This is one of the difficult aspects of rapid response team, the presepsin could be a useful marker in diagnosing septic shock. Presepsin is helpful not only as a diagnosis of septic shock, but also as an indicator for predicting mortality along with lactic acid and SOFA scores. When AKI is accompanied, it may be measured higher than patients without AKI, it is necessary to pay attention to interpretation and consider other markers to diagnosis septic shock or prediction of mortality.”
Thanks reviewers of Journal of Clinical Medicine.
Sincerely yours,
Jee Hee Kim
Department of Anesthesiology, National Cancer Center
323 Ilsan-ro, Ilsandong-gu, Goyang-si 410-769, Gyeonggi-do, Republic of Korea
E-mail: [email protected]

Round 2
Reviewer 1 Report
The authors have addressed moreover all questions raised in the previous version of the manuscript
Reviewer 2 Report
The manuscript prepared by Lee et al has been improved